# Application of Mesenchymal Stem Cell Therapy and Inner Ear Regeneration for Hearing Loss: A Review

**DOI:** 10.3390/ijms21165764

**Published:** 2020-08-11

**Authors:** Sho Kanzaki, Masashi Toyoda, Akihiro Umezawa, Kaoru Ogawa

**Affiliations:** 1Department of Otolaryngology, Keio University School of Medicine, 35 Shinanomachi, Shinjuku, Tokyo 160-8582, Japan; ogawak@keio.jp; 2Research Team for Geriatric Medicine, Tokyo Metropolitan Institute of Gerontology, 35-2 Sakae-cho, Itabashi, Tokyo 173-0015, Japan; mtoyoda@tmig.or.jp; 3National Center for Child Health and Development, 2-10-1 Okura, Setagaya, Tokyo 157-8535, Japan; umezawa@1985.jukuin.keio.ac.jp

**Keywords:** inner ear, regeneration, mesenchymal stem cells, imaging, hearing loss

## Abstract

Inner and middle ear disorders are the leading cause of hearing loss, and are said to be among the greatest risk factors of dementia. The use of regenerative medicine for the treatment of inner ear disorders may offer a potential alternative to cochlear implants for hearing recovery. In this paper, we reviewed recent research and clinical applications in middle and inner ear regeneration and cell therapy. Recently, the mechanism of inner ear regeneration has gradually been elucidated. “Inner ear stem cells,” which may be considered the precursors of various cells in the inner ear, have been discovered in the cochlea and vestibule. Research indicates that cells such as hair cells, neurons, and spiral ligaments may form promising targets for inner ear regenerative therapies by the transplantation of stem cells, including mesenchymal stem cells. In addition, it is necessary to develop tests for the clinical monitoring of cell transplantation. Real-time imaging techniques and hearing rehabilitation techniques are also being investigated, and cell therapy has found clinical application in cochlear implant techniques.

## 1. Introduction

The incidence of patients with sensorineural hearing loss, including age-related hearing loss (presbycusis), has increased. Moreover, evidence linking hearing loss to heightened risks of cognitive function impairment, such as dementia [1], has raised concerns over the issue and resulted in increased research into new therapies for inner ear disorders, including inner ear regenerative medicine. In this paper, we review recent research and clinical applications in inner ear regeneration and cell therapy.

Hearing loss is classified into two types: conduction hearing loss and sensorineural hearing loss. Conductive hearing loss is an abnormality of the middle ear (tympanic membrane and auditory ossicles), which affects the ability to convey sound vibrations, whereas sensorineural hearing loss is due to inner ear disorder [2].

Chronic otitis media (COM) is the primary cause of conductive hearing loss. This condition involves perforation of the tympanic membrane and erosion of the ossicles caused by repeated infections. The tympanic membrane is regenerated using the fascia or perichondrium. However, hearing improvement surgery may be ineffective if the tympanic membrane lacks stem cells [3]. In cases involving bone erosion, other ossicles or cartilage may be used as substitutes in hearing improvement surgery. Mesenchymal stem cells (MSCs) can also be useful to treat conductive hearing loss [4].

The etiologies of sensorineural hearing loss disorders include aging, genetics, acoustic trauma, drug-induced hearing loss, infections, immune disorders, endolymphatic hydrops (Meniere’s disease), and sudden sensorineural hearing loss [5]. Vulnerability of the inner ear causes severe inner ear disorders in many patients. It is exceptionally difficult to regenerate the mammalian inner ear functionally and anatomically once it has been injured. Consequently, there are few effective available treatments for inner ear disorders, and functional recovery can be expected in very few cases [5]. 

Cochlear implants have been able to restore certain degree of auditory function in patients with severe hearing loss; however, this treatment is insufficient because those cells are not regenerated. However, research into alternative regenerative therapies began at the end of the 20th century, and mechanisms of inner ear regeneration have gradually been elucidated [6].

The inner ear has three components: the scala vestibuli (SV), scala media (SM), and scala tympani (ST), and is composed of hair cells or sensory cells, spiral ligaments (including fibrocytes), and stria vascularis, which regulates cochlear potential in the SM, along with primary auditory neurons or spiral ganglion neurons [2]. In the auditory system, sounds are transmitted through the external auditory canal, causing the eardrum to vibrate. These vibrations pass through the middle ear to the inner ear. The inner ear is filled with fluid, which passes vibrations on to sensory cells called hair cells [2]. Hair cells actively vibrate, resulting in oscillations that cause the ion channels to open. The hair cells depolarize, and current is transmitted to the primary auditory neurons, known as spiral neurons [2]. The current finally reaches the auditory nerves, brain stem, thalamus, and auditory cortex [7]. Research into regenerative approaches have resulted in the elucidation of some factors required for the regeneration of hair cells, mainly based on an improved understanding of the mechanism of inner ear development. The induction of differentiation in endogenous stem cells present in the inner ear and inner ear stem cell transplantation of hair cells, neurons, and spiral ligament fibrocytes may be possible. Recently, “inner ear stem cells,” which may be the precursors of various cells in the inner ear, have been discovered in the cochlea (hearing organ) and vestibule (balance organ).

Mesenchymal stem cells (MSCs) are found throughout the body, including bone marrow, fat, and skin, and the properties of MSCs differ slightly depending on their location [8]. Although MSCs are originally defined as cells that can differentiate into fat, bone, and cartilage, they can also differentiate into certain other tissue cells such as hepatocytes or neurons.

There is a high risk of rejection with the transplantation of an organ composed of donor induced pluripotent stem cells (iPSCs) [9]. However, MSCs have an immunomodulatory ability that aids transplantation and reduces the risk of rejection. However, many concerns have been raised over the immunogenic potential of induced pluripotent stem cells (iPSCs) [10]. A recent study demonstrated that iPSCs have similar immunogenic and more potent immunomodulatory properties than those of bone marrow-derived stromal cells in vitro [11].

The difference between MSCs and induced pluripotent stem cells (iPSCs) is that treatment with MSCs has the advantage of reducing the risk of tumor formation [12], which is a problem with iPSCs. MSCs have already been commercialized. In fact, cell therapy containing MSCs is used as a treatment for graft-versus-host disease after bone marrow transplantation. Furthermore, the treatment of spinal cord injury using bone marrow-derived MSCs will be clinically available soon. This treatment promotes the regeneration of neurons in the spinal cord by culturing MSCs collected from the patient’s own bone marrow fluid and returning them to the body. Cell therapy for the inner ear currently requires local injection into the inner ear. A recent study has shown that the transtympanic administration of bone marrow-derived mesenchymal stem cells (BM-MSCs) causes oxidative stress or inflammatory response in immunocompetent cochlea in rats [13]. Hence, research into the administration methods of transplanted cells is also clinically important.

In basic hearing and regeneration research, we can perform histological and functional (audiological) animal studies (i.e., preclinical studies) using objective hearing function tests based on the auditory brainstem response (ABR), which is extracted from the ongoing electrical activity in the brain in response to sound stimuli. Although animal ears are tiny organs, these studies are advantageous and provide samples for molecular biological studies.

## 2. Regeneration of the Inner Ear

The inner ear is comprised of sensory epithelium including hair cells, supporting cells that support hair cells anatomically and physiologically, spiral ganglion neurons, both cochlear spiral ligaments and stria vascularis to maintain homeostasis of the inner ear. Unlike birds, in the mammalian inner ear, hair cells cannot regenerate after damage without difficulty. No regenerative therapy to treat irreversible inner ear damage in humans has yet become available. Thus, hearing aids and cochlear implants (electric stimulation to spiral ganglion neurons instead of the hair cells) are the only clinical options. Vertigo and dizziness are symptoms of disorders in equilibrium organs such as vestibular hair cells, and, unlike auditory organs, vestibular organs can partially regenerate in mammals. 

Although the mechanism of regeneration of hair cells remains unknown, the transcription factor Atoh1 [14], POU domain factor Pou4f3 [15], and Zinc finger Gfi1 [16], which are required to differentiate progenitor cells into hair cells, have been discovered. A recent study showed that Six1, Atoh1, Pou4f3, and Gfi1 can convert fibroblasts into hair cells similar to those in the ear [17].

The differentiation of feeder cells into hair cells and signaling pathways has also been elucidated. Hair cells can be regenerated by locally injecting a gamma secretase inhibitor that inhibits Notch signaling pathway activity and differentiates hair cells from progenitors (i.e., supporting cells) to improve hearing by 10 decibels [18]. However, this level of improvement is not sufficient. In addition, the administration is within 3 days, because Hes5 mRNA expression increased and gradually decreased to reach the pre-noise level by 3 days after noise exposure after the noise exposure. New hair cell generation results from increased levels of the bHLH transcription factor Atoh1 in supporting cells in response to the inhibition of Notch signaling in mice with acoustic trauma, in which the hair cells have been damaged. This therapeutic effect and improvement range are small; therefore, treatment should aim to enhance further hearing improvements for clinical applications [18].

MicroRNA (miRNA) miR-96, miR-182, and miR-183 [19], which are a class of highly conserved endogenous non-coding small RNAs, regulate stem cell proliferation and differentiation [20]. An increased expression of miRNA-182, but neither miRNA-96 nor miRNA-183, in BM-MSCs could lead to higher expression levels in some hair cell markers. The cell morphology did not change. However, miRNA-182 could have an important function in hair cell differentiation via the upregulation of SOX2, POU4F3, and ATOH1 to promote a hair cell’s fate. 

Using light-emitting diodes (LEDs) with a wavelength of 630 nm, the differentiation of hair cell-like cells derived from mouse embryonic stem cells (ESCs) was coupled with the overexpression of reactive oxygen species (ROS). A transcriptome analysis revealed the factors correlated with the effect of PBM on the formation of otic organoids [21].

Ito et al. demonstrated that neural stem cells can be engrafted into the sensory epithelium of a model animal ear [22]. Recently, significant progress has led to the identification of hair cells, spiral ganglion cells, spiral ligaments, and stria vascularis as promising targets for inner ear regenerative medicine. Cell therapies involving the replacement of lost and injured cells with newly transplanted cells and the injection of genes and drugs to regenerate the remaining cells of the inner ear represent a promising area of research [23]. Previous findings suggest that stem cells can engraft in the inner ear environment; thus, they can be used for inner ear transplantation. One issue requiring future resolution involves the potential of engrafted stem cells to differentiate into various cells in the inner ear. Several unknown signals or growth factors are known to initiate the differentiation of stem cells into cells with distinct properties (e.g., hair cells and neurons). Although the detailed mechanism underlying induction is unknown, stem cells were shown to differentiate into inner ear hair cells [24]. When stem cells were transplanted into an inner ear under a conditioned micro-environment, they partly differentiated into inner ear cells and partly into hair cells, which might have contributed to the regeneration of the sensory epithelium. When cells were injected directly from the lateral cochlea wall, the engraftment of these cells was observed in the cochlea, vestibular sensory epithelium, and spiral ganglia [24]. The migration of neural stem cells into the injured inner ear tissue under this condition showed superior results than the transplantation into the uninjured inner ear [25]. In this review, we have classified three therapeutic strategies: stem cell therapy for hair cells and neurons prior to cochlear implantation, gene or drug therapy to stimulate supporting cells, and drug screening for inner ear regeneration, as shown in Table 1. In the future, regenerative therapies combining stem cells, genes, and drug therapies will also be performed.

We also expect that drug screening will be applied to stem cells. Drugs that would facilitate hearing improvements, protection, and inner ear regeneration could be selected efficiently, if screened using large amounts of sensory cells or neurons induced from stem cells prepared in vitro, without expensive and time-consuming animal experiments. However, no studies have reported drug screening for inner ear regeneration this must be elucidated in the future. In this article, we discuss the potential of stem cell (i.e., MSC) therapy. 

## 3. Potential of Cell Therapy for Sensory Cells Using MSCs 

Ideally, transplanted progenitors would replace both lost hair cells and neurons in the inner ear. To achieve hearing recovery during inner ear regeneration, transplanted cells must survive in the inner ear, engraft in anatomically and functionally correct positions, and differentiate correctly into target cells, such as hair cells (without turning cancerous), whereas neural regeneration is required if spiral ganglion neurons (primary neurons) do not remain and differentiated cells must synergize correctly with the auditory nerve. MSCs also would solve these issues.

The characteristics of inner ear stem cells are not defined. However, otic progenitor cells (OPCs) express several specific markers, such as PAX2, PAX8, SOX2, OTX1, FOXG1, GATA3, and TBX1 [26,27]. The OPCs have been isolated from the sensory epithelium of the utricle, an adult mouse vestibular sensory organ [28,29], and have also been derived from sensory epithelial cells [30]. Sensory epithelial cells, including hair cells and supporting cells, may revert to stem cell-like cells in inner ear disorders. 

Oshima et al. [26] improved the protocol presented by Li [31] to yield a purely in vitro method without mouse ESC and iPSC engraftment. The cells were differentiated into hair cells through the progenitor cells. Both ESC- and iPSC-derived OPC cultures responded to utricle stromal cells by organizing into clusters. Regenerative medicine can be practiced using these cells. Embryonic stem cells (ESCs) and iPSCs can functionally and morphologically regenerate into hair cells [26]. However, MSC-derived hair cells have not been functionally tested, and there are no reports comparing iPSC- and MSC-derived hair cells. MSCs are more suitable for mesenchymal cells in the inner ear, such as cochlear fibrocytes, than for sensory hair cells.

MSCs are advantageous because they have been administered to humans in certain studies [29]. In these pilot studies, the safety of an intravenous injection and transplantation of autologous MSCs has been verified in patients with sensorineural hearing loss; however, there is insufficient improvement in hearing [29]. 

Although MSCs have a limited differentiation capacity, these can be differentiated into neural stem cells and transplanted into inner ear fibrocytes, which represent significant advantages. The ability to apply MSCs to experimental autoimmune hearing loss represents another advantage [32]. In an earlier study, human adipose tissue-derived stem cells (ASCs) decreased the antigen-specific Th1/Th17 cell populations, and induced the production of the anti-inflammatory cytokine interleukin-10 using splenocytes [32].

MSC engraftment was observed in the stria vascularis [33], spiral ligament fibrocytes [34], and ST fibrocytes [35] when MSCs were administered. There were some differences in the engraftment sites between the reports. In one report, MSCs (bone marrow stromal cells) transplanted into the spiral ligament of the cochlea; however, certain grafted cells expressed neural or glial cell markers, indicating their ability to differentiate into neuronal or glial cells, respectively [34]. Due to 3-nitropropionic acid, a significant recovery in the thresholds of auditory brainstem responses after MSC transplantation, via the posterior semicircular canal, was only found at 40 kHz in a mild degeneration model of the spiral ligament [36]. MSCs enhanced the regeneration and maintenance of fibrocytes in damaged spiral ligaments (SLs), leading to the partial restoration of cochlear function in rats [36]. Hematopoietic stem cells (HSC) were also transplanted into spiral ligament cells [37] (Figure 1). The spiral ligament maintains the inner ear homeostasis and is involved in inner ear electromotive force. Potassium ions are taken up by the stria vascularis from the spiral ligament in the lateral wall and secreted into the endolymph. Degeneration of the spiral ligament is observed during the development of the presbycusis. The degeneration of fibrocytes in the spiral ligament preceded the loss of hair cells and neurons. Furthermore, it presents a useful treatment target for intractable inner ear disorders, such as Meniere’s disease and sudden sensorineural hearing loss.

MSCs can be transformed into different cell types. However, MSCs might engraft more easily after differentiation into neural stem cells in vitro, followed by the administration to the inner ear. The best methods for transplanting differentiated cells into tissues have not yet been determined, and it remains to be elucidated which pathological conditions of the inner-ear-transplanted cells are more likely to survive. According to the characteristics of MSCs, they may be useful for transplantation into mesenchymal tissues such as spiral ligaments. The type of stem cell is also selected according to the type of target inner ear cell. In addition, neurotrophic support molecules secreted from MSCs prevent hair cell degeneration. Mononuclear cells derived from HSCs also prevent ischemia-induced damage to the inner hair cells [38] (Figure 1).

## 4. Cell Therapy for Inner Ear Spiral Ganglion Neuron Degeneration Using MSCs 

Spiral ganglion neurons are primary auditory neurons. Damage or loss of inner ear hair cells follows neural degeneration, and the number of remaining spiral ganglion neurons has been shown to be associated with the outcome of cochlear implant treatment [46]. An electrode was inserted into the cochlea of a guinea pig model of drug-induced deafness as a substitute for a cochlear implant, and electrical stimulation was performed. Compared with electrical stimulation alone or glial-derived neurotrophic factor (GDNF) gene therapy alone, the combination of GDNF gene therapy and electrical stimulation significantly prevented neuronal degeneration [47]. The introduction of a brain-derived neurotrophic factor (BDNF) transgene improved the threshold of electrically evoked auditory brainstem responses (eABR), which can be induced by electrical rather than sound stimuli. The results suggest an improvement in neuronal function [48]. Therefore, maintaining the number of spiral ganglion neurons may have important clinical implications. Trophic support derived from MSCs improve the therapeutic outcome of the cochlear implant by preventing degeneration of neurone, similar to prevention of hair cells as described [48].

To differentiate neurone from bone marrow MSCs, overexpression of Neurog1 developed glutamatergic neurons (spiral ganglion neurons) even though it was not sufficiently maintained [43]. In this report, the authors quantified the upregulation of transcription factors expressed by developing primary auditory neurons, such as BRN3a (POU4F1) and GATA3, after inducing the expression of Neurog-1. Moreover, the expression of the receptor NTRK2 was induced by treatment with BDNF, its specific ligand. The induction of the expression of the vesicular glutamate transporter 1 expression was identified at the gene and protein levels. NeuroD1 did not appear to sufficiently induce and maintain neuronal differentiation. The induction of neuronal differentiation caused by the overexpression of Neurog1 initiated important steps in the development of glutamatergic neurons, such as spiral ganglion neurons. However, this overexpression was not sufficient to maintain the glutamatergic spiral ganglion neuron-like phenotype [43]. A protocol for neural differentiation from MSCs remains to be established. Cell therapy may improve the performance of cochlear implants by recruiting depleted neurons [42]. Undifferentiated mouse MSCs survived in the modiolus (auditory nerve) of the gerbil cochlea [49]. Adult human MSCs obtained from nasal tissue have been used to repair the spiral ganglion loss in experimentally lesioned cochlear cultures from neonatal rats. These stem cells restored the population of spiral ganglion neurons via both direct neuronal differentiation and secondary effects on endogenous cells [42] (Figure 1).

Cell therapy could feasibly regenerate spiral ganglion neurons. Reports have described improved hearing loss following implantation of human ESC-derived neural progenitor cells into the inner ears of the gerbils, in which the spiral ganglion neurons had been destroyed [50]. This is the first report describing the use of cell transplantation to improve hearing loss. However, clinically, injuries rarely involve only spiral ganglion cells; in most cases, the hair cells are also injured. Accordingly, its use is still far from clinical application. Moreover, trophic factors may be secreted by transplanted neural stem cells [51], and may effectively promote the survival of the surrounding neurons.

Other research has explored the development of trophic support from bone marrow-derived mononuclear cells (BM-MNCs) on electrodes for cochlear implants [44]. Those cells can release growth factors and anti-inflammatory cytokines with neuroprotective effects and can thus improve immune responses to cochlear implant devices, which are recognized as foreign bodies. 

## 5. Three-Dimensional (3D) Organoid Model of the Inner Ear (Cochlea and Vestibular Organ)

The cochlea has an extremely complicated structure (with one inner hair cell and three outer hair cells elegantly arrayed). It is very difficult to reorganize structure in damaged cochlea. Future studies on organoids should focus on disease models, such as those of developmental, genetic and degenerative disorders and cancer. Several studies have established an extremely valuable disease model using stem cells. Organoids can be formed directly from human cells, thereby facilitating a more effective identification of drug efficacy and toxicity. The use of an organoid can significantly reduce the use of animals in preclinical studies.

Several factors must be considered when establishing an organoid. Pax2 [52] or Fbxo2 [53] can induce organoid formation to model otic induction, which is a pivotal developmental event when the preplacodal tissue adopts otic fate.

In 3D organoid derived from mouse ESCs, the preplacodal and otic placode ectoderm was induced by the inhibition of BMP signaling and the addition of fibroblast growth factor 2 (FGF2). Supplementation with BDNF and neurotrophin-3 (NT-3) was used for further maturation in the vitro differentiation of SGNs with a bipolar morphology and functional excitability [54].

To promote epithelialization, mouse ESCs are aggregated in a medium containing extracellular matrix (ECM) proteins. Within the first 14 days, accurately timed protein degradation and small molecular treatments can sequentially induce the expression of epithelia that represent the embryonic non-neural ectoderm, preplacodal ectoderm and otic vesicle epithelia in mice. These tissues develop into cysts with a pseudostratified epithelium containing inner ear hair cells and supporting the cells of sensory-like neurons that generate synapse-like structures with the derived hair cells after 16–20 days [55].

Similarly, the ECM is crucial in mimicking a stem cell niche in vitro and in driving stem cells toward the three dimensions formation. Technological developments have led to the investigation of biomaterials similar to the native ECM [56]. 

The application of Lgr5-positive cochlear progenitors in a 3D organoid for testing several candidate drugs and gene silencing and overexpression is a good in vitro tool for the analysis of progenitor cell manipulation and hair cell differentiation [57].

Several protocols have been established to help direct iPSCs into the hair cells and neurons, which have several properties similar to those of their native counterparts. The efficiency, reproducibility and scalability of these protocols are enhanced by incorporating knowledge on inner ear development [58,59].

Generating otic organoids generated from human iPSCs using 3D culture system can form hair cells bearing stereociliary bundles with active mechanosensory ion channels. These cells have morphological characteristics similar to those of their in vivo counterparts during the embryonic development of cochlear and vestibular organs. Moreover, they present with electrophysiological activity detected via single-cell patch clamping [58]. In addition, the use of human disease models in vitro via the genetic manipulation of iPSCs is feasible [59]. For example, inner ear organoids with single-cell RNA sequencing were used to investigate the role of type II transmembrane protease 3 (Tmprss3). Tmprss3-deficient hair cells had reduced numbers of channels and lower expressions of genes encoding calcium ion-binding proteins. This result indicated a disruption in intracellular homeostasis. In addition to the endoplasmic reticulum (ER) of cells, a proteolytically active TMPRSS3 was detected in the cell membranes [60]. 

Organoid cultures achieve a rapid, robust, and reproducible induction of sensory cells and neurons with the potential to be used in the future for inner ear regeneration and for understanding the pathology of genetic hearing loss.

## 6. Administration of Stem Cells into the Inner Ear 

The inner ear is located inside the tympanic membrane and can be approached comparatively easily with an incision in the tympanic membrane. Although the inner ear is surrounded by bone, this area is filled with endolymph and perilymph fluids. With considerable advantage, administered stem cells can easily diffuse in these fluids [61]. The inner ear is a functional organ converting sound waves (vibrations) into electric currents, and large vibrations or mechanical damage to the inner ear may lead to inner ear damage, which requires micro-perfusion into the inner ear [62]. Anatomically, the inner ear is divided into the SM, ST and SV, and the method of transplant surgery should be selected according to the space in which the target site is located [63]. The ionic composition of lymph fluid differs between ST, SV and SM, and the environment of the inner ear changes, depending on the location [63]. Furthermore, the mixing of perilymph and endolymph fluids from different spaces can damage the inner ear function. Hair cells can be targeted by injecting cells directly into the cochlea (via the SM) and a route via the endolymphatic sac [64], which controls the production and absorption of endolymph fluid. An endolymphatic sac surgery is performed in cases of severe-grade Meniere’s disease [65], and would be applied to stem cell regeneration medicine.

Neuron targeting occurs using administration via the ST. This relatively easy process is the same as that used to insert electrodes for cochlear implants. All administration methods could cause inner ear disorders. However, the effect of hearing deterioration appears to be less dependent on the ST aspect of gene administration to the inner ear [66]. The cochlea is connected by a vestibular organ. In animal experiments, however, stem cells are often administered via the semicircular canals [36,40]. This prevents hearing damage when the cells are administered.

Another option involves the intravenous administration of MSCs, which was verified as safe in a human study [29]. However, this technique remains questionable because the stem cells can penetrate the blood-labyrinth barrier. Recent studies showed that the transtympanic administration of rodent BM-MSCs was based on cochlear function, and the occurrence of any adverse effects in the auditory system was assessed using a non-immunocompromised rat model. The transtympanic administration of BM-MSCs has no significant effect on the hearing thresholds, as determined based on ABR and distortion product otoacoustic emissions (DPOAE), which can be used to measure outer hair cell function. Histopathological examination did not reveal recruitment of inflammatory leukocytes and edema in the cochlea of MSCs administered in rats [67].

Further animal experiments are needed to elucidate these concerns.

## 7. Imaging for Tracking Transplanted Cells

Clinically, it is necessary to examine how transplanted cells survive in the inner human ear. To elucidate the condition of cells transplanted in the inner ear, inner ears should be removed and examined in individual animals. Better imaging techniques are now being developed. A contrast agent containing iron (known as superparamagnetic iron oxide (SPIO)) was incorporated into bone marrow MSCs-derived neural stem cells, and the cells were administered to the inner ear of a guinea pig. By measuring the imaging effect using a 1.5 Tesla MRI, which is in clinical use, it was possible to confirm that the stem cells were engrafted in the inner ear [35]. In this study, the change in signal intensity at the transplanted site was consistent with the histological decrease in the number of labelled cells, suggesting that cochlear macrophages contributed to the decrease in the number of transplanted cells until 4 weeks after transplantation. Histological findings demonstrated that the transplanted cells were localized in the organs of Corti and modiolous (i.e., cochlear nerve) outside the ST, which was the transplantation site. This observation suggests the possibility of transplanted cell migration and engraftment and the monitoring of transplanted SPIO-labeled cells in the inner ear via MRI. The detection period was limited and determined to be within 2 weeks in this study. The histological findings of all specimens obtained in this study demonstrated no evidence of surgical damage other than fenestration to ST (the target site), confirming that the transplant was performed appropriately. We counted a very small number of cells at several sites, as described by previous reports. Some clinical questions remain unanswered; for example, how the macrophages phagocytosed the transplanted cells that appeared in the cochlea and whether MRI signal intensity changes are associated with inner-ear disorders. Regarding the dynamics of macrophages, a study examined SPIO-labeled cells transplanted into mouse thigh muscle. The researchers confirmed that transplanted MSCs underwent apoptosis and disappeared approximately 4–8 weeks after transplantation. As the ST, which is filled with perilymph fluid, was the transplanted site, the cochlear perilymph metabolism must also be considered. MRI studies of patients with severe hearing loss did not reveal any change in signal relative to that observed in normal counterparts, suggesting that inner ear disorders do not affect MRI signals. In future studies, MRI scanners with a minimum magnetic force of 1.5 T and maximum forces as high as 3.0 T should be used. In animal experiments, in vivo imaging should be performed, along with a system that can directly visualize living cells using the luciferin/luciferase luminescence reaction [35]. Real-time imaging is the key to the clinical application of stem cell transplantation and engraftment confirmation.

## 8. Auditory Rehabilitation after Stem Cell Transplantation

The auditory network must be reconstructed after inner ear regeneration. Although the above-mentioned animal experiments show improvement in hearing after treatment, hearing sound is not clinically sufficient, as, in humans, understanding language is also important. Hence, auditory rehabilitation is essential as a method to supplement regeneration techniques. Hearing rehabilitation has already been performed after the administration of hearing aids [68] and cochlear implants [69], and researchers are working on applying these findings. Auditory nerve network reconstruction occurs after cochlear implant surgery, and brain plasticity is present even in adults [70]. Therefore, patients with a recent onset of hearing loss and cochlear implant surgery may be better candidates for stem cell therapy because the appropriate networks may remain. Currently, patients with severe hearing loss may reacquire hearing if the neural network constructed by the cochlear implant is used and transplanted new hair cells have formed connections with the cochlear (auditory) nerves. It is also necessary to observe the state of the neural network in the ear in the living body using real-time imaging.

## 9. Addressing Clinical Problems Associated with Inner Ear Regenerative Medicine

So far, we have described the basic research on inner ear regeneration and the potential for regenerative medicine. A major issue in regenerative medicine for hearing in the clinical setting is how to understand the various deafness conditions in each case. For example, the policy of regenerative medicine differs depending on the following scenarios as below and in Table 1:
Scenario (1): when only hair cells are lost but supporting cells and neurons remain.Scenario (2): when all the cells in the inner ear, including all the hair cells, the supporting cells, and all the neurons, are lost.


In Scenario (1), a hair cell differentiation inducer such as drug and gene therapy can be applied. However, Scenario (2) will require cell therapy without indication of a differentiation inducer. If there is a long gap between the onset of severe hearing loss and its treatment, there is a high possibility that sensory epithelial cells and several types of cells in the inner ear have been completely lost and cannot be regenerated, which may be an indication for cell therapy. If we were able to determine which cells remain in the inner ear, treatments making full use of the patient’s cells become possible. At present, the absence of such a test is a challenge facing clinical applications.

Cochlear implants are indicated in cases of severe hearing loss for which hearing aids are not sufficient [71]. In patients with cochlear implant indications, treatment typically involves inserting a cochlear implant with the aim of preserving hearing, and then using a combination of cell therapy and drug administration. This is probably the closest that the emerging field of inner ear regenerative medicine has come to clinical application.

The in vitro production of target cells derived from a patient with genetic hearing loss would help to elucidate the pathology of genetic hearing loss and lead to the development of treatment methods without the collection of pathological cells from the inner ear of a patient. However, in certain pathological conditions, the disease is not always at the cellular level. Therefore, it may not be possible to clarify all pathologies, because the inner ear organs and their surrounding tissues cannot be reproduced in this type of analysis. For example, an analysis may be insufficient for deafness caused by a synaptic abnormality and diseases associated with inner ear malformation.

The clinical application of MSCs remains challenging. For example, when administering MSCs, the stage of cell differentiation and whether MSCs themselves or neural stem cells (or precursors) can be more easily engrafted remain to be determined. The reorganization of a neural network after MSC engraftment also requires further exploration.

## 10. Conclusions

A review of regenerative medicine using MSCs for the auditory system, especially, inner ear cochlea was presented. The application of cell therapy to inner ear regeneration was demonstrated, and the possibility of the regeneration of hair cells, spiral ganglion neurons, stria vascularis and spiral ligaments was described. We have also shown that spiral ganglion neuron regeneration could improve the outcome of patients with cochlear implants, and this is likely the closest that the emerging field of regenerative inner ear medicine has come to a clinical application. Real-time imaging techniques and hearing rehabilitation techniques are also under investigation, and cell therapies have been applied clinically in cochlear implant techniques.

## Figures and Tables

**Figure 1 ijms-21-05764-f001:**
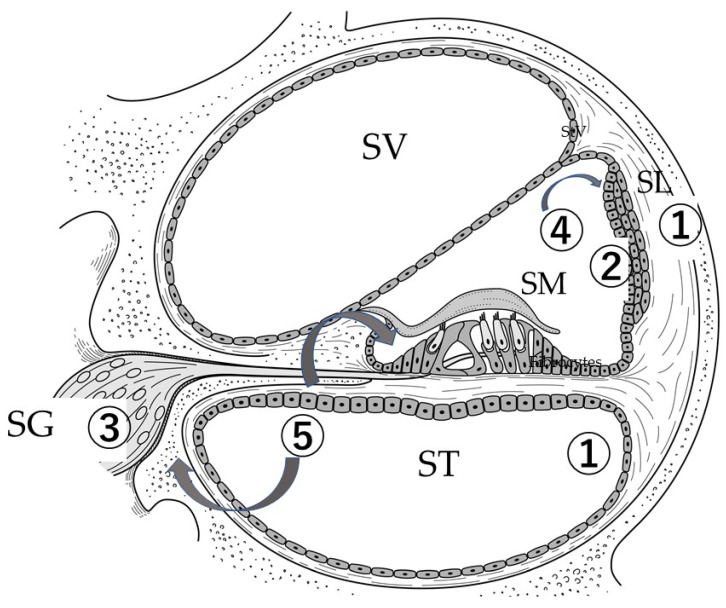
Transplantation of mesenchymal stem cells into the inner ear. Differentiation from mesenchymal stem cells into: (1) cochlear fibrocytes (spiral ligament), hematopoietic stem cells (HSCs) [37], mesenchymal stem cells (MSCs) [36,39], MSCs in scala tympani (ST) [35] and MSCs in ST of only young mice [40]; (2) stria vascularis, MSCs (IV) [33]; (3) spiral ganglion neurons, MSC (IV) [41], MSCs from nasal turbinates in vitro [42], MSCs into glutamatergic neurons [43], bone marrow-derived mononuclear cell (BM-MNC) sheet-coated electrodes in humans at cochlear implantation [44]; (4) immune cells, macrophage with Iba1 [45], adipose tissue-derived stem cells (ASCs) in autoimmune inner ear disease [32]; (5) neurotrophic support, mononuclear cells from HSCs; prevention of ischemia-induced damage to the inner hair cells [38]. SV: Scala vestibuli; SG: Spiral ganglion; SM: Scala media; ST: Scala tympani; SL: Spiral ligament.

**Table 1 ijms-21-05764-t001:** Classification of inner ear regeneration therapy.

(1) Cell therapy using MSCs, iPSCs, and ESCs Transplantation of hair or supporting cells Transplantation of neurons in cases of cochlear implantation for patients with severe hearing loss Neurotrophic support from transplanted cells for degenerative neurons before cochlear implantation
(2) Drug screening for inner ear regeneration using stem cells (3) No report
(4) Gene or drug therapy (5) ● Hair cell regeneration (Atoh1 or Six1, Atoh1, Pou4f3, and Gfi1 or Notch inhibitors) treatment if supporting cells remain ● Neurotrophic support treatment for degenerative neurons before cochlear implantation

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
