# Peer review of "Application of Mesenchymal Stem Cell Therapy and Inner Ear Regeneration for Hearing Loss: A Review"

_ijms, 2020, doi:10.3390/ijms21165764_

Round 1

Reviewer 1 Report

The authors responded in details to all reviewer questions/suggestions. No further modifications are necessary. I am pleased to recommend this manuscript for publication in IJMS. 

Reviewer 2 Report

The paper can be published in its present form. Implemented data and changes improved its quality.

Only few minor typing mistakes were found: line 117- double  'after the noise exposure' and line 115- double dot.

Reviewer 3 Report

This is  a nice review of the current state of the art for inner ear regeneration. This is a very broad topic and the key areas are well addressed.

A thorough proof-reading is necessary for content and usage.

This manuscript is a resubmission of an earlier submission. The following is a list of the peer review reports and author responses from that submission.

Round 1

Reviewer 1 Report

Even though there is no original information, this is a well written article to review and summarize the use of mesenchymal cell in inner ear regeneration. 

Author Response

Dear Reviewr1, Thank you  for your comments. we summarized regeneration therapy of inner ear  and stem cells (MSCs) .

Reviewer 2 Report

A concise presentation that clearly describes the latest research in the field of using MSCs for the auditory system.

It is suitable for publication in its present form. 

I found only few minor mistakes:

The Table 1 was partially lost probably during converting to pdf

Line 160 delete ‘e’

Line 362 - double parenthesis-  (ST))

Author Response

Dear Reviewer 2

Thank you for your comments. I corrected mistakes you pointed out.

1) The Table 1 was partially lost probably during converting to pdf.

>I inserted Table 1.

2)Line 160 delete ‘e’  

> I deleted ‘e’

3) Line 362 - double parenthesis-  (ST))

> (ST)

Reviewer 3 Report

The review may be of interest to the readers of IJMS, but needs more work to clarify all the issues raised in the paper.

  1. Appropriate references in the Introduction are required to give credit to the authors/contributors to the field
  2.  Introduction cont. "The current finally reaches the auditory nerves, brain stem, and auditory brain
    - Between brain stem and auditory cortex there is thalamus. Also, „auditory brain”? Do you mean "auditory cortex" ?
  3.  Introduction cont. „The difference between MSCs and induced pluripotent stem cells (iPSCs) is that MSCs have an immunomodulatory function. … When transplanting an organ made from donor iPSCs, there is a high risk of rejection; however, MSCs have immunomodulatory ability, which aids
    transplantation and reduces the risk of rejection.”- in fact, iPSC have been reported to display immunomodulatory properties (e.g. Schnabel, Lauren V et al. “Induced pluripotent stem cells have similar immunogenic and more potent immunomodulatory properties compared with bone marrow-derived stromal cells
    in vitro.” Regenerative medicine vol. 9,5 (2014): 621-35. doi:10.2217/rme.14.29)
  4. It would be of benefit to add more details regarding the characteristics of IESC (i.e. inner ear stem cells) and their specific (if any) markers. 
  5.  It would be also desirable to indicate potential/reported differences in the differentiation potential of iPSC and MSC to generate hair cells 
  6. Regeneration of inner ear: „Although the mechanism of regeneration of hair cells remains unknown, the transcription factor
    Atoh1, which is required to differentiate progenitor cells into hair cells, has been discovered [2]" - Regarding the hair cells gene expression profile, besides Atoh1, there is also
    Pou4f3, Gfi1, and miRNA-183.
  7. Regeneration of inner ear cont.: „The differentiation of feeder cells into hair cells and signaling pathways have also been elucidated. Hair cells can be regenerated with a gamma secretase inhibitor that inhibits the
    Notch signaling pathway and differentiate hair cells to improve hearing by 10 decibel, which is
    not enough recovery" - please add more info re: "gamma secretase inhibitor" and this study
  8. Regeneration of inner ear cont.: „Embryonic stem cells (ESCs) and iPSCs can be functionally and morphologically regenerated into hair cells via inner ear stem cells from iPSCs [10]" - ESCs and iPSCs in this study were cultured with utricle stromal cells - please comment!
  9. Regeneration of inner ear cont.: „In one report, MSCs transplanted into the spiral ligament of the cochlea, but some grafted cells expressed a neural or glial cell marker, indicating their ability" - please indicate what type/origin of MSC were used in this study
  10. Fig. 1 is unclear and confusing, please verify
  11. „To differentiate neuron from bone marrow MSCs, overexpression of Neurog1 developed glutamatergic neurons (spiral ganglion neurons) even though it was not sufficiently maintained [23]. The protocol of neural differentiation from MSCs should be established" - some info re:
    neurog1?

Author Response

Dear Reviewer 3

Thank you for your comments.

  1. Appropriate references in the Introduction are required to give credit to the authors/contributors to the field

>We have included several references in the Introduction.

  1. Introduction cont. "The current finally reaches the auditory nerves, brain stem, and auditory brain”
    - Between brain stem and auditory cortex there is thalamus. Also, „auditory brain”? Do you mean "auditory cortex" ?

>Auditory cortex is correct. I have changed “auditory brain” to “auditory cortex” and inserted “thalamus” where appropriate.

  1. Introduction cont. „The difference between MSCs and induced pluripotent stem cells (iPSCs) is that MSCs have an immunomodulatory function. … When transplanting an organ made from donor iPSCs, there is a high risk of rejection; however, MSCs have immunomodulatory ability, which aids transplantation and reduces the risk of rejection.”- in fact, iPSC have been reported to display immunomodulatory properties (e.g. Schnabel, Lauren V et al. “Induced pluripotent stem cells have similar immunogenic and more potent immunomodulatory properties compared with bone marrow-derived stromal cells
    in vitro.” Regenerative medicine vol. 9,5 (2014): 621-35. doi:10.2217/rme.14.29)

https://www.ncbi.nlm.nih.gov/pubmed/25372080

>Although I have read the publication suggested by the reviewer, this topic appears to remain controversial. I have revised the following sentences.

However, many concerns have been raised over the immunogenic potential of induced pluripotent stem cells (iPSCs) [6],[7]. A recent study demonstrated that iPSCs have similar immunogenic and more potent immunomodulatory properties than those of bone marrow-derived stromal cells in vitro [8].

The difference between MSCs and induced pluripotent stem cells (iPSCs) is that...

  1. It would be of benefit to add more details regarding the characteristics of IESC (i.e. inner ear stem cells) and their specific (if any) markers. 

> The characteristics of inner ear stem cells are not defined. However, otic progenitor cells (OPCs) express several specific markers, such as PAX2, PAX8, SOX2, OTX1, FOXG1, GATA3, and TBX1 [17] [18].The OPCs have been isolated from the sensory epithelium of the utricle, an adult mouse vestibular sensory organ [19], and have also been derived from sensory epithelial cells [20]. Sensory epithelial cells, including hair cells and supporting cells, may revert to stem cell-like cells in inner ear disorders.

It would be also desirable to indicate potential/reported differences in the differentiation potential of iPSC and MSC to generate hair cells.

> Oshima et al. [17] improved the protocol presented by Li [19] to yield a purely in vitro method without mouse ESC and iPSC engraftment. The cells were differentiated into hair cells through the progenitor cells. Both ESC- and iPSC-derived OPC cultures responded to utricle stromal cells by organizing into clusters. Regenerative medicine can be practiced using these cells. Embryonic stem cells (ESCs) and iPSCs can functionally and morphologically regenerate into hair cells [17]. However, MSC-derived hair cells have not been functionally tested, and there are no reports comparing iPSC- and MSC-derived hair cells. MSCs are more suitable for mesenchymal cells in the inner ear, such as cochlear fibrocytes, than for sensory hair cells.

  1. Regeneration of inner ear: „Although the mechanism of regeneration of hair cells remains unknown, the transcription factor Atoh1, which is required to differentiate progenitor cells into hair cells, has been discovered [2]" - Regarding the hair cells gene expression profile, besides Atoh1, there is also Pou4f3, Gfi1, and miRNA-183.

>Although the mechanism of regeneration of hair cells remains unknown, the transcription factor Atoh1, POU domain factor Pou4f3, Zinc finger Gfi1, and miRNA-183, which are required to differentiate progenitor cells into hair cells, has been discovered [9] [10] [11].

Regeneration of inner ear cont.: „The differentiation of feeder cells into hair cells and signaling pathways have also been elucidated. Hair cells can be regenerated with a gamma secretase inhibitor that inhibits the Notch signaling pathway and differentiate hair cells to improve hearing by 10 decibel, which is not enough recovery" - please add more info re: "gamma secretase inhibitor" and this study

>The differentiation of feeder cells into hair cells and signaling pathways have also been elucidated. Hair cells can be regenerated by locally injecting a gamma secretase inhibitor that inhibits Notch signaling pathway activity and differentiates hair cells from progenitors (i.e., supporting cells) to improve hearing by 10 decibels. However, this level of improvement is not sufficient. New hair cell generation results from increased levels of the bHLH transcription factor Atoh1 in supporting cells in response to the inhibition of Notch signaling in mice with acoustic trauma, in which the hair cells have been damaged.

  1. Regeneration of inner ear cont.: „Embryonic stem cells (ESCs) and iPSCs can be functionally and morphologically regenerated into hair cells via inner ear stem cells from iPSCs [10]" - ESCs and iPSCs in this study were cultured with utricle stromal cells - please comment!

>The characteristics of inner ear stem cells are not defined. However, otic progenitor cells (OPCs) express several specific markers, such as PAX2, PAX8, SOX2, OTX1, FOXG1, GATA3, and TBX1 [17] [18].The OPCs have been isolated from the sensory epithelium of the utricle, an adult mouse vestibular sensory organ [19], and have also been derived from sensory epithelial cells [20]. Sensory epithelial cells, including hair cells and supporting cells, may revert to stem cell-like cells in inner ear disorders.

Oshima et al. [17] improved the protocol presented by Li [19] to yield a purely in vitro method without mouse ESC and iPSC engraftment. The cells were differentiated into hair cells through the progenitor cells. Both ESC- and iPSC-derived OPC cultures responded to utricle stromal cells by organizing into clusters. Regenerative medicine can be practiced using these cells. Embryonic stem cells (ESCs) and iPSCs can functionally and morphologically regenerate into hair cells [17]. However, MSC-derived hair cells have not been functionally tested, and there are no reports comparing iPSC- and MSC-derived hair cells. MSCs are more suitable for mesenchymal cells in the inner ear, such as cochlear fibrocytes, than for sensory hair cells.

Regeneration of inner ear cont.: „In one report, MSCs transplanted into the spiral ligament of the cochlea, but some grafted cells expressed a neural or glial cell marker, indicating their ability" - please indicate what type/origin of MSC were used in this study

>We have added the following sentence: “

In one report, MSCs (bone marrow stromal cells) transplanted into the spiral ligament of the cochlea; however, certain grafted cells expressed neural or glial cell markers, indicating their ability to differentiate into neuronal or glial cells, respectively [24].”

  1. figure1 is unclear and confusing, please verify

>We have clarified figure 1.

  1. „To differentiate neuron from bone marrow MSCs, overexpression of Neurog1 developed glutamatergic neurons (spiral ganglion neurons) even though it was not sufficiently maintained [23]. The protocol of neural differentiation from MSCs should be established" - some info re:neurog1? 

>According to your suggestion, we have inserted some additional information on neurog1.

In this report, the authors quantified the upregulation of transcription factors expressed by developing primary auditory neurons, such as BRN3a (POU4F1) and GATA3, after inducing the expression of Neurog-1. Moreover, the expression of the receptor NTRK2 was induced by treatment with BDNF, its specific ligand. The induction of the expression of the vesicular glutamate transporter 1 expression was identified at the gene and protein levels. NeuroD1 did not appear to sufficiently induce and maintain neuronal differentiation. The induction of neuronal differentiation caused by the overexpression of Neurog1 initiated important steps in the development of glutamatergic neurons, such as spiral ganglion neurons. However, this overexpression was not sufficient to maintain the glutamatergic spiral ganglion neuron-like phenotype[32]. A protocol for neural differentiation from MSCs remains to be established.